# Effect of Heat–Moisture Treatment on the Physicochemical Properties, Structure, Morphology, and Starch Digestibility of Highland Barley (*Hordeum vulgare* L. var. *nudum* Hook. f) Flour

**DOI:** 10.3390/foods11213511

**Published:** 2022-11-04

**Authors:** Yiming Lv, Sen Ma, Jingyao Yan, Binghua Sun, Xiaoxi Wang

**Affiliations:** College of Food Science and Engineering, Henan University of Technology, Zhengzhou 450001, China

**Keywords:** heat–moisture treatment, highland barley, starch structure

## Abstract

This study modified native highland barley (HB) flour by heat–moisture treatment (HMT) at different temperatures (90, 110, and 130 °C) and moisture contents (15%, 25%, and 35%). The effects of the treatment on the pasting, thermal, rheological, structural, and morphological properties of the native and HMT HB flour were evaluated. The results showed that HMT at 90 °C and 25% moisture content induced the highest pasting viscosity (3626–5147 cPa) and final viscosity (3734–5384 cPa). In all conditions HMT increased gelatinization temperature (*T_o_*, 55.77–73.72 °C; *T_p_*, 60.47–80.69 °C; *T_c_*, 66.16–91.71 °C) but decreased gelatinization enthalpy (6.41–0.43 J/g) in the HMT HB flour compared with that in the native HB flour. The HB flour treated at 15% moisture content had a higher storage modulus and loss modulus than native HB flour, indicating that HMT (moisture content, 15%, 25%, and 35%) favored the strengthening of the HB flour gels. X-ray diffraction and Fourier-transform infrared spectroscopy results showed that HMT HB flour retained the characteristics of an A-type crystal structure with an increased orderly structure of starch, while the relative crystallinity could be increased from 28.52% to 41.32%. The aggregation of starch granules and the denaturation of proteins were observed after HMT, with additional breakage of the starch granule surface as the moisture content increased. HMT could increase the resistant starch content from 24.77% to 33.40%, but it also led to an increase in the rapidly digestible starch content to 85.30% with the increase in moisture content and heating temperature. These results might promote the application of HMT technology in modifying HB flour.

## 1. Introduction

Highland barley (*Hordeum vulgare* L. var. *nudum* Hook. f) is mainly produced in the Qinghai Tibet Plateau and is the main food crop of the Tibetan people. It is widely used in the food industry in the form of alcoholic beverages, pasta products, and baked goods [1]. Consumption of HB helps facilitate bowel movement, prevent colon cancer, improve immunity, reduce blood lipid and cholesterol levels, and reduce the risk of diabetes and hyperlipidemia [2]. Therefore, HB can be used as an important component of functional food for preventing chronic diseases. Unfortunately, HB tastes rough and does not form dough easily, and starch as the main ingredient may have a negative effect on blood glucose levels. With the increasing demand for healthy food in recent years, it is essential to modify HB flour to improve the palatability and nutrition of HB products.

Thermal treatment is a common processing method used in the food industry and is considered fast, simple, and environmentally friendly. Various thermal technologies exist for modifying starch and flour, such as the HMT, annealing, pre-gelatinization, roasting, flash explosion, microwave heating, and superheated steam treatment. However, roast and flash explosions may increase the amount of damaged starch [3]. The depth of microwave penetration during microwave modification can have a different effect on the starch, thus reducing the efficiency of the modification [4]. The complexity of the superheated steam processing equipment leads to high maintenance costs. Vapor condensation can occur on the surface of the sample during the initial stages of superheated steam processing, leading to possible changes in moisture content [5]. Among these thermal modification methods, HMT is not only simple to pre-treat raw materials, but also does not require complex heating equipment, and the process is relatively short. Therefore, HMT has low costs and ease of operation, making it easier to scale up in the food industry. HMT is a physical method used to modify starch and flour under water content of less than 35% and high temperature [6]. Its main principle is that heat energy can give starch molecular energy, which intensifies the movement of molecular chains. On the contrary, it can promote the migration of water molecules among the layered, crystalline, and amorphous regions of starch granules and generate new hydrogen bonds [7]. These changes affect the original aggregation and chain structure of the starch, altering the pasting properties and digestibility of the starch. Research has revealed that the physicochemical properties of heat–moisture treated starch might change, such as the change in the granule structure, the increase in swelling power, and the improvement in the rheological properties of paste [8,9]. Moreover, HMT is considered to be particularly promising for the production of food additives, bakery products, frozen food, and so on [9]. HMT barley flour can be applied in bread matrices; it positively influences the technical function and nutritional parameters of the bread and exhibits a similar appearance, texture, and flavor to wheat flour bread [10]. The shelf life of biscuits has been extended with added HMT rice flour [11]. Pasta with HMT corn starch has a good expansion, minimal cooking time, and firm texture (e.g., processing and evaluation of HMT amaranth starch noodles, which is an inclusive comparison with corn starch noodles) [12]. HMT flour also has good thermal stability and shear resistance for applications in sauces, cans, and other products [13].

The objective of this study was to investigate the influence of HMT on the properties of HB flour. In this study, X-ray diffraction (XRD) and Fourier-transform infrared spectroscopy (FTIR) were used to examine the structure of starch and proteins to better understand the effects of different temperatures and moisture content of HMT on HB flour. Scanning electron microscopy (SEM) and confocal laser scanning microscopy (CLSM) were used to observe the morphology of the flour granules. Additionally, the pasting, thermal, rheological, and digestibility properties of HB flour were determined, which are useful to understand the effect of HMT on the properties of HB flour.

## 2. Materials and Methods

### 2.1. Materials

HB flour was obtained from Qinghai New Green Health Food Co., Ltd. (Xining, China). The chromatographic grade was required for potassium bromide (KBr). Other reagents used were of analytical grade.

### 2.2. Preparation of Samples

An appropriate amount of distilled water was sprayed evenly and slowly on the surface of the HB flour (200 g, db), and constant stirring was carried out to avoid uneven mixing. The moisture content of the HB flour was adjusted to 15%, 25%, and 35%, respectively, and then equilibrated at a temperature of 4 °C for 24 h in sealed polyethylene bags. The samples were then placed into screw-capped glass containers (500 mL) and heated at 90 °C, 110 °C, and 130 °C for 2 h in a hot-air oven. After HMT, the HB flour was dried at 40 °C for 24 h. The obtained HB flour samples were marked as S_n_ (native HB flour), S_90−15_, S_90−25_, S_90−35_, S_110−15_, S_110−25_, S_110−35_, S_130−15_, S_130−25_, and S_130−35_ (S_A−B_, where A represents the heating temperature and B represents the moisture content).

### 2.3. Pasting Properties

The pasting properties of starch were determined using a rapid visco analyzer (RVA4500; Perten Instruments, Macquarie Park, NSW, Australia) by the method suggested by Feng et al. [14]. We took 25 g of distilled water into the sample cylinder of the rapid viscosimeter. Then, a 3.5 g sample of flour (with 14% moisture) was weighed and added to the sample cylinder, and stirred with a stirrer for proper mixing of the sample flour. The test was conducted using the following procedure: the mixture was stirred quickly for 10 s (960 rpm) to properly mix the sample. The sample was kept at 50 °C for 1 min, heated to 95 °C at a constant speed for 3.42 min, kept at 95 °C for 2.7 min, cooled to 50 °C at a constant speed for 3.88 min after the heat preservation stage, and finally kept at 50 °C for 2 min. The stirring speed in the test stage was 160 rpm.

### 2.4. Thermal Properties

The thermal properties of HB flour were determined using a differential scanning calorimeter (DSC Q20; TA Instruments, New Castle, DE, USA) by the method described by Ma et al. [15]. We took 2.0 mg (db) of the HB flour sample in an aluminum plate and added 6 μL of distilled water. The sealed aluminum plate was kept at 25 °C for 24 h for proper mixing. The properly mixed sample was then kept in the DSC sample room; the heating process was conducted from 25 to 120 °C at a rate of 10 °C/min. The experimental data were analyzed using the TA Instruments Universal Analysis 2000 Version 4.5 software.

### 2.5. Rheological Characteristics

The rheological properties of HB flour under different HMTs were measured using HAAKE RS6000 rheometer (Thermo Fisher Haake, Karlsruhe, Germany) by the method described by Wang et al. [16]. We prepared the sample to be tested in 10% (*w*/*w*) suspension, placed the suspension on the magnetic stirrer for 1 h, and then took 1 mL of the suspension on the rheometer disk. The rotor was PP35Ti with a gap of 1 mm. A proper amount of silicone oil was applied around the plate and then the plate was covered to prevent the sample from losing water during heating. The steady-state flow test was conducted at a shear rate of 0.1–100 s^−1^, and the dynamic viscoelastic test was conducted at an angular frequency of 0.1–100 rad/s at 1% strain within the linear viscoelastic region.

### 2.6. Fourier-Transform Infrared Spectroscopy

HB flour and KBr (chromatographic grade) were fully ground in agate mortar at the ratio of 1:99 and then pressed into transparent thin slices. The thin slices were scanned 32 times continuously using an FTIR spectrometer (Nicolet FT-IR; Thermo Fisher Scientific Co., Waltham, MA, USA) in the wavelength range of 4000−400 cm^−1^. The OMNIC 8.2 and PeakFit 4.12 software were used to process the experimental data.

### 2.7. X-ray Diffraction

HB flour was tested using an X-ray diffractometer (Minifle × 600; Rigaku Denki Co., Tokyo, Japan) operated at 40 kV and 40 mA, at a scanning angular range of 5° to 40°, which was a function of 0.02° (2θ) at a scanning speed of 2°/min. The diffraction pattern was processed using the MDI Jade 6 software (Materials Data Inc., Livermore, CA, USA) to analyze the crystal type and relative crystallinity.

### 2.8. Scanning Electron Microscopy

A scanning electron microscope (Quanta 250FEG, FEI, Hillsboro, OR, USA), under an accelerating voltage of 10.0 kV, was used to examine the granular morphology of HB flour at 1000× magnification. The HB flour was fixed on the sample table with conductive tape, and the microstructure of HB was observed under the accelerating voltage of 10 kV after spraying gold.

### 2.9. Confocal Laser Scanning Microscopy

Starch and protein were stained with fluorescein isothiocyanate (FITC), rhodamine B, respectively. About 60 mg of HB flour was dissolved in 700 μL of deionized water, then 30 μL of FITC (1 mg/mL) and 30 μL of rhodamine B (1 mg/mL) were added. Then, the sample was placed in the dark for 30 min and observed using CLSM (FV3000, Olympus Co., Tokyo, Japan). The excitation wavelengths of FITC and rhodamine B were 488 and 543 nm, respectively.

### 2.10. In Vitro Digestibility of Starch

The in vitro digestive characteristics of the samples were determined by the method described by Gu et al. with minor modifications [17]. Each flour sample (200 mg of starch, db) and 10 glass balls were added to 15 mL of acetate–sodium acetate buffer solution (0.1 mol/L, pH 5.2) and then preheated for 10 min at 37 °C in water. Subsequently, 5 mL of the enzyme solution (290 U/mL porcine pancreatic α-amylase and 15 U/mL amyloglucosidase) was added and incubated at 37 °C with a shaking rate of 300 rpm. The hydrolysate samples (1 mL) were taken after 0, 20, and 120 min, and the enzyme was inactivated using a boiling water bath. After centrifugation (10,000 rpm, 5 min), 200 µL of the supernatant was pipetted and mixed with 800 µL of distilled water. We added 200 µL of the diluted solution to 400 µL of DNS and then boiled in a water bath for 5 min. The absorbance was measured at 540 nm. Tests were run in triplicate. The rapidly digestible starch (RDS), slowly digestible starch (SDS), and resistant starch (RS) contents were calculated as follows:RDS (%) = (G_20_ − G_0_) × 0.9/TS × 100
SDS (%) = (G_120_ − G_20_) × 0.9/TS × 100
RS (%) = 100 − RDS − SDS
where G_0_, G_20_, and G_120_ represent the amount of glucose released within 0, 20, and 120 min, respectively; and G_0_ is the amount of glucose in the sample before hydrolysis. TS represents the total starch content. The total starch content of HB flour was analyzed using α-amylase method and Total Starch Assay Kit (K-TSTA, Megazyme International Ltd., Wicklow, Ireland) [18].

### 2.11. Statistical Analysis

All data were analyzed at least in duplicate. The SPSS Statistical 20.0 software was used to conduct one-way analysis of variance and the test for statistical significance (Duncan, *p* < 0.05) for all experimental data obtained, and the results are shown as the mean ± standard deviation. The analysis of all the images was conducted using Origin 8.5.

## 3. Results and Discussion

### 3.1. Pasting Properties of Native and HMT HB Flour

The pasting properties of native and HMT HB flour determined using RVA are presented in Table 1. After HMT, the pasting temperature increased significantly (*p* < 0.05), indicating the disintegration and rearrangement of the double-helix structure of starch and the development of crystalline structures [19]. The interaction between starch chains and the interaction between starch and proteins also led to an increase in the gelatinization temperature [20]. The HB flour samples S_90−15_, S_90−25_, and S_110−15_ had higher peak viscosity than that of S_n_. On the contrary, S_90−35_, S_110−25_, S_110−35_, S_130−15_, S_130−25_, and S_130−35_ had lower peak viscosity than that of S_n_. This result indicated that HMT conditions of 90 °C and 15% moisture content caused the disordering of starch. The decrease in peak viscosity might be related to the decrease in the swelling capacity of the starch granules, which was caused by the decomposition and rearrangement of starch after HMT [21]. At 110 °C and 130 °C, the pasting temperature increased, and the peak viscosity, breakdown, and final viscosity declined with the increase in moisture level. The peak viscosity of S_130−15_ did not show a statistically significant difference (*p* > 0.05) probably due to the lower moisture content during HMT. Pasting occurred only on the surface of the starch granules, and a hard shell was formed, preventing water from entering the interior of the starch granules, thus inhibiting the gelatinization of starch [22]. The breakdown level of S_90−35_, S_110−25_, S_110−35_, S_130−25_, and S_130−35_ was lower than that of S_n_. This result might be attributed to the fact that adequate water tended to promote the rearrangement of starch chains, which strengthened the amorphous region of starch and made the breakdown of starch granules difficult [23]. The gelatinization of S_90−15_ stayed in the initial stage due to moisture or temperature restrictions, when mainly amylose was leached from the starch granules [9]. The retrogradation of amylose led to an increase in final viscosity and setback viscosity during the cooling phase of the RVA test. HMT could reduce the leaching of amylose in starch [24]. The decrease in the level of breakdown and setback indicated that the strength of internal chemical bonds of starch increased after HMT, and starch needed more energy decomposition, which confirmed more interaction [25].

### 3.2. Thermal Properties

The gelatinization parameters of native and HMT HB flour are presented in Table 2. HMT had increased onset temperature (*T_o_*), peak temperature (*T_p_*), and conclusion temperature (*T_c_*) but decreased gelatinization enthalpy (Δ*H*) than the native counterpart. These changes in gelatinization temperature with increasing moisture content of the HB flour system at the same heating temperature were mainly because the increase in moisture content facilitated the enhancement of the mobility and flexibility of the crystalline and amorphous regions of the starch, promoting intermolecular interactions to produce an ordered structure with higher thermal stability [26,27]. The increase in *T_o_*, *T_p_*, and *T_c_* of HB flour after HMT could be due to the interactions between amylose and amylose, amylose and amylopectin, and amylose and fat molecules, which limited the flexibility of the internal amorphous structure of starch and hindered its swelling [26]. Consequently, higher thermal energy was required to swell the amorphous regions of the starch. When the HB flour system was under the same moisture conditions, the more thermal energy brought in with the increase in heating temperature caused damage to the crystalline regions, thus increasing the thermal stability of the starch. Another reason for the increase in gelatinization temperature when the system moisture content was 15% might be that the proteins covering the surface of the starch during HMT blocked the channels on the surface of the starch granules, limiting the penetration of water molecules into the starch crystals and affecting the swelling of the starch granules. Δ*H* represents the quality and quantity of crystals in starch granules, which reflects the energy required to destroy the molecular interaction in the starch structure during gelatinization [28]. When the system moisture content was 15%, Δ*H* did not change significantly with the increasing heating temperature; however, this reduction in Δ*H* was pronounced when the system moisture content was 35%. The decrease in gelatinization enthalpy indicated that HMT promoted water molecules to enter the crystalline region from the amorphous region and destroyed the hydrogen bond in the starch, leading to the disintegration of the double helix in the crystalline region and the disruption of the amorphous region [29]. This effect could be exacerbated by an increase in heating temperature. However, HMT caused rearrangement at the starch molecular level, increased the interaction between starch molecular chains, and improved the quality of available double-helix sequences [30].

### 3.3. Rheological Properties

The rheological properties of native and HMT HB flour were measured using the steady shear test and frequency sweep test. The storage modulus (G′) of all samples was higher than the loss modulus (G″), indicating that these samples had solid-like characteristics, except S_110−35_, S_130−25_, and S1_30−35_ (Figure 1). In particular, the G′ and G″ of samples S_90−15_, S_90−25_, S_110−15_, and S_130−15_ were higher than those of native flour. This result suggested that HMT favored the strengthening of HB flour gels under low-moisture content. With the increase in temperature and moisture content in HMT, G″ and G′ decreased gradually, which indicated that starch granules had a higher degree of gelatinization and hindered the formation of a strong gel structure. The loss tangent was defined by tan *δ* = G″/G′. Tan *δ* of samples S_110−35_, S_130−25_, and S_130−35_ were greater than one, which indicated that the flour paste was mainly viscous and its fluidity was enhanced. Under the synergistic effect of thermal energy and water, the crystalline region of starch granules melted, starch granules broke, molecular fluidity increased, and the interaction between chains weakened [31]. Thermal denaturation and aggregation of proteins might also affect this result. The high temperature and high moisture content led to the cross-linking of disulfide bonds, hydrogen bonds, and other noncovalent interactions, thus changing the viscoelasticity [32].

The apparent viscosities of HB flour paste under different HMT conditions decreased sharply first with the increase in the shear rate, and then tended to be flat, showing typical shear thinning and non-Newtonian behaviors (Figure 1). Generally, a starch paste is a thixotropic fluid with a three-dimensional network structure, which is formed by the interaction of hydrogen bonds between molecules. Nevertheless, the interaction of hydrogen bonds is fragile, so it is easy to break when sheared, and the gel structure is gradually destroyed. The structure of amylose and amylopectin in starch granules is destroyed by the synergistic effect of water and heat. Some amylose formed a new structure on the surface of swollen starch granules after precipitation, which hindered the gelatinization of starch granules [22]. When the water content was 35%, a large amount of water accelerated the movement of molecules, weakened the entanglement between starch molecular chains in flour paste, and led to a decrease in the apparent viscosity of flour paste. We found that the apparent viscosity of S_90−15_ was higher than the apparent viscosity of the native HB flour paste. We hypothesized that, with the increase in humidity and heat treatment temperature, the thermal movement of molecules accelerated, the chances of amylose movement in starch increased, the molecular chains entangled with each other, and the apparent viscosity of starch paste also increased.

### 3.4. XRD Analysis

The XRDs of HB flour were obtained to examine the effect of HMT on the crystalline structure (Figure 2). As shown in Figure 2, all samples displayed a typical A-type crystallinity with strong diffraction pattern peaks at 2θ of 14.98°, 17.94°, 19.88°, and 23.04°. The samples S_90−15_, S_90−25_, S_110−15_, and S_130−15_ showed similar A-type crystal structures as the native HB flour, indicating that the crystal type did not change. However, the peak at 19.88° of samples S_90−35_, S_110−25_, S_110−35_, S_130−25_, and S_130−35_ was higher than that of native HB flour, showing a V-type crystal structure, which represented the interaction between amylose and lipids due to complex formation [33]. The more crystals in starch, the higher the intensity of the diffraction peak. The diffraction peak intensities of samples S_90−15_, S_90−25_, S_110−15_, and S_110−25_ were higher than those of native HB flour, which indicated that new crystals were formed in the starch after HMT. The decrease in the diffraction peak intensities of samples S_130−25_ and S_130−35_ at 14.98° indicated the loss of crystal arrangement. This might be due to the damage of the hydrogen bond, leading to the displacement of adjacent double-helix and incomplete parallel rearrangement [34]. Also, the relative crystallinity of HMT HB flour increased (Table 3). This result might be attributed to the limited moisture content. The relative crystallinity of HMT HB increased, which might be due to the interplay of several factors such as an increase in the chain movement, promotion of the interaction between amylose and amylopectin to form new crystals, and disintegration and rearrangement of the double helix of starch granules, making the microcrystalline area of starch granules more perfect [35]. The water and thermal energy could promote the formation of a double-helix structure [36]. However, the relative crystallinity of samples S_130−25_ and S_130−35_ began to decrease compared with that of the other HMT HB flour. The excessive heat and moisture during HMT caused a reduction in crystallinity. The result suggested that the relative crystallinity in starch granules was destroyed after HMT [37]. The partial or complete gelatinization of starch and the change in the double-helix structure during heat treatment might destroy starch granules and change the orientation of starch crystallites. This might be the main reason for the decrease in crystallinity of high-moisture starch [37]. Besides, high-moisture starch was sensitive to regular repetitive double-helix segments (long-range order structures) of XRD, but less sensitive to irregularly ordered segments (short-range order structures).

### 3.5. FTIR Spectrum Analysis

FTIR is another analytical technique used to characterize the ordered structure of starch. The FTIR spectra of native and HMT HB flour are presented in Figure 3, and the bands’ absorption area ratios at 1047/1022 cm^−1^ are summarized in Table 4. The native and the HMT HB flour had similar peaks, indicating that no new groups were generated after HMT. The 800–1200 cm^−1^ bands were considered as the fingerprints of starch, which reflected the stretching vibration of starch C–C, C–OH, and C–H bonds [38]. The bands at 1047 cm^−1^ were characterized by the structural features of the starch crystalline regions, corresponding to the ordered structure in the starch. The bands at 1022 cm^−1^ showed the structural characteristics of the amorphous region of starch, which belonged to the random curly structure of starch macromolecules. The absorbance ratio at 1047 cm^−1^/1022 cm^−1^ (R_1047/1022_) reflected the ratio of short-range ordered structures in starch. The higher the ratio, the more the short-range ordered structures in starch. The intensity of HMT HB flour at 1047 cm^−1^ and 1022 cm^−1^ decreased compared with that of native HB flour. The decrease in the absorption peak of the HMT HB flour might be attributed to the strengthening of the stretching vibration of the C=O, C–O–C, or C–C bond [39]. After HMT, the R_1047/1022_ of the treated samples was higher compared with that of the native HB flour. This result might be related to the retrogradation of the starch. During starch retrogradation, a part of the double helix structures re-formed by starch chains were arranged in order to form a long-range ordered structure, while another part of the new double helix structures were arranged in order to form a short-range ordered structure. FTIR is very sensitive to the determination of this short-range order structure [40]. The result showed that the external region structure of the HMT flour became more ordered. Similar results were also reported in the studies by Afzal Ali and Lu [41,42]. However, the R_1047/1022_ of sample S_130−15_ was lower than that of the native HB flour. This result might be attributed to the fact that HMT led to partial gelatinization of starch, which might destroy the double-helix structures and crystal domains. Moreover, when the sample was heated at high temperatures, the water evaporated and the water content of HB flour decreased, which was not conducive to starch retrogradation.

The secondary structure of proteins is usually studied via amide I corresponding band (1600–1700 cm^−1^). We determined the proportion of the secondary structure content of HB protein (Table 5). The HB protein mainly consisted of β-sheets and β-turns. As shown in Table 5, HMT led to a decrease in the proportion of β-turns and an increase in the proportion of β-sheets. Besides, the content of the α-helix structure varied during HMT. The increase in the number of β-turns might be at the expense of the decrease in the number of β-turns, and this change was irreversible at temperatures above 45 °C and high-moisture content [43]. During high-temperature heating, the molecular chain of HB protein extended, rearranged, and aggregated, leading to the formation of the β-turn structure [44].

### 3.6. SEM and CLSM Analyses on HB Flour

The granule morphology of native and HMT HB flour is shown in Figure 4. The starch granules and the conjunction of protein and starch were observed. HB starch and protein are shown in green and red, respectively, in the CLSM image. The non-starch components of HB flour, including proteins and barn, were distributed around starch granules or adhered to the surface of starch granules. The surface of the native HB starch granules was smooth, and its shape was mainly large oval (A-type starch granule), disc-like (A-type starch granule), and small round (B-type starch granule). The morphology of starch granules in HB flour heated at 15% moisture content did not change, non-starch components adhered to the surface of starch granules, and starch granules also adhered to each other to form small lumps. It might be related to the leached amylose, which resulted in the association of granules with each other [38]. The surface appearance results of samples S_90−15_, S_110−15_, and S_130−15_ showed no apparent change, which indicated that the changes caused by HMT under low-moisture conditions mainly occurred inside the HB starch granules. With the increase in HMT moisture content, the starch granules were destroyed more obviously, and the aggregation of starch granules was observed, which might be related to partial gelatinization. Excessive water in the HMT process caused the swelling and melting of HB starch granules, resulting in the interconnection of starch granules. During drying and crushing, the morphology of starch granules was also destroyed. Heating intensified the swelling and melting of starch granules, and finally made starch granules lose their original identity [45]. The surface of heat–moisture treated HB starch granules was rough. This might be due to the fact that water entered the amorphous area of starch granules, starch granules swelled after being heated, and the surface of starch granules collapsed after cooling. It was also possible that the existence of weak tissue in the internal structure of starch granules was influenced by a thermal force exerted by HMT to recombine amylose and amylopectin within the weaker regions, resulting in the formation of more dense amorphous sites [38,46]. Besides, proteins were denatured and aggregated by heat, and the higher the temperature, the more pronounced the change.

### 3.7. Starch Digestibility Analysis of HB Flour

The contents of RDS, SDS, and RS for native and HMT HB flour are presented in Table 6. The contents of RDS, SDS, and RS in S_n_ were 60.35%, 14.87%, and 24.77%, respectively. HB flour treated at 90 °C or 15% moisture content showed no significant change in the RDS content. The RDS content could reach 85.30% when both heating temperature and moisture content were increased. The SDS content increased with increasing moisture content at heating temperatures of 90 °C and 110 °C; however, it increased and then decreased with increasing moisture content at 130 °C. The RS content of HB treated at 15% moisture content increased to 33.40%. The formation of RS after HMT was mainly due to the interactions between starch chains and the rearrangements of starch molecules that could weaken the sensitivity to digestive enzymes [27]. HMT also promoted a densely packed aggregation structure of starch that prevented digestive enzymes from binding to starch granules. Furthermore, under low-moisture conditions, the proteins were denatured by heat and adhered to the surface of the starch granules, reducing the contact site for digestive enzymes [47]. The increase in RDS and SDS content was due to the disruption of the ordered structure of the starch molecular chains and the weakening of the interactions between the molecular chains, making the starch granules susceptible to attack by digestive enzymes [48,49]. When the heating temperature and moisture content of the HMT were increased, the thermal energy could promote the entry of water into the starch granules, resulting in the breaking of hydrogen bonds or double helices in the crystalline and amorphous regions of the starch and the full extension of the starch chains [50]. These changes made it easier for starch to bind to digestive enzymes.

## 4. Conclusions

The pasting, thermal, rheological, and digestibility properties of HB flour after HMT changed significantly compared with that of native HB flour. These changes indicated the interactions between starch chains, interactions between starch and protein, or changes in the internal structure of the starch. The synergistic effect of heat and water caused the starch molecular chains to degrade and the double-helix structure to deconvolve during the HMT process. However, it also promoted the free movement of the starch molecular chains; the starch molecular chains were rearranged and reoriented, making the double-helix structure and the crystalline structure more perfect and increasing the ordered structure of the amorphous crystalline region and raising the RC of HMT HB flour to 41.32%. These changes ultimately improved the thermal stability of HMT HB flour. After the changes observed at 15% moisture content of HB flour, no significant changes were found in the morphology of the starch granules. The viscosity of the low-moisture content of the flour pastes increased. However, as the moisture content increased, the granules aggregated with each other, with tight junctions between the granules, forming agglomerates and damaging the starch surface. Protein aggregation mainly occurs with an increase in temperature. After the aforementioned changes, HMT induced changes in starch structure easily, such that HB starch bound to digestive enzymes, leading to an increase in RDS content. HB flour treated with low-moisture content could not easily bind to digestive enzymes, leading to an increase in the RS content. In general, HMT broadened the scope of the application of HB flour in the food industry, the effect of different HMT conditions on HB flour was variable, and in particular, the effect of moisture differences was the most significant. The conditions of HMT could be adjusted to obtain the desired HB flour properties.

## Figures and Tables

**Figure 1 foods-11-03511-f001:**
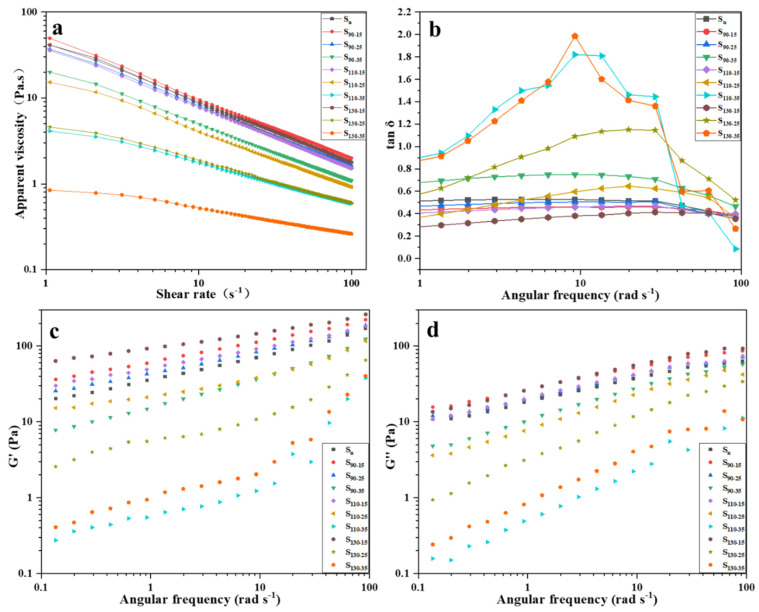
Rheological properties of native and HMT HB flours. (**a**) Apparent viscosity of HB flour gel; (**b**) tan δ; (**c**) storage modulus; (**d**) lose modulus. S_n_ indicates native HB flour; S_90−15_, S_90−25_, S_90−35_ indicate that the heating temperature of HB flour is 90 °C; S_110−15_, S_110−25_, S_110−35_ indicate that the heating temperature of HB flour is 110 °C; S_130−15_, S_130−25_, S_130−35_ indicate that the heating temperature of HB flour is 130 °C; S_90−15_, S_110−15_, S_130−15_ indicate that the moisture content of HB flour is 15%; S_90−25_, S_110−25_, S_130−25_ indicate that the moisture content of HB flour is 25%; S_90−35_, S_110−35_, S_130−35_ indicate that the moisture content of HB flour is 35%; G′, storage modulus; G″, loss modulus; tan *δ*, loss tangent.

**Figure 2 foods-11-03511-f002:**
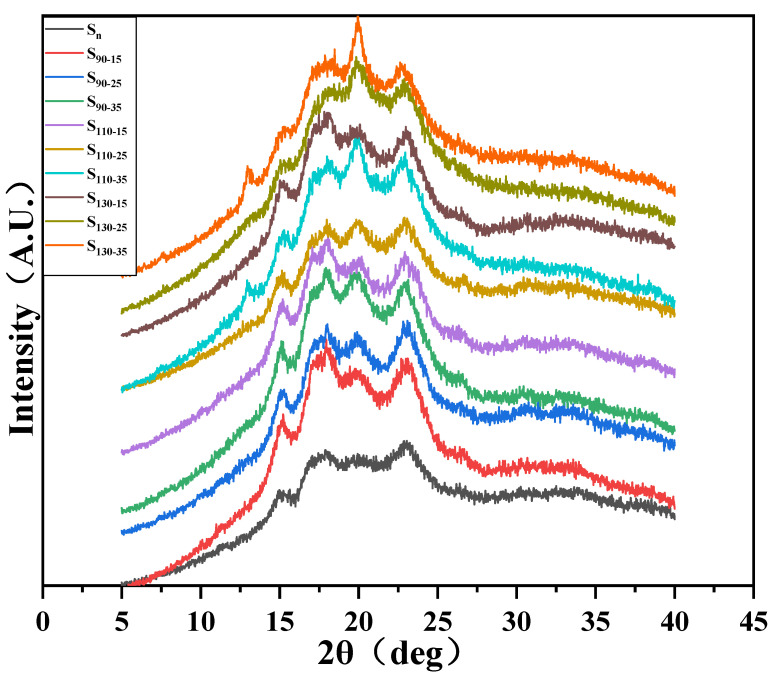
XRD pattern of native and HMT HB flours. S_n_ indicates native HB flour; S_90−15_, S_90−25_, S_90−35_ indicate that the heating temperature of HB flour is 90 °C; S_110−15_, S_110−25_, S_110−35_ indicate that the heating temperature of HB flour is 110 °C; S_130−15_, S_130−25_, S_130−35_ indicate that the heating temperature of HB flour is 130 °C; S_90−15_, S_110−15_, S_130−15_ indicate that the moisture content of HB flour is 15%; S_90−25_, S_110−25_, S_130−25_ indicate that the moisture content of HB flour is 25%; S_90−35_, S_110−35_, S_130−35_ indicate that the moisture content of HB flour is 35%.

**Figure 3 foods-11-03511-f003:**
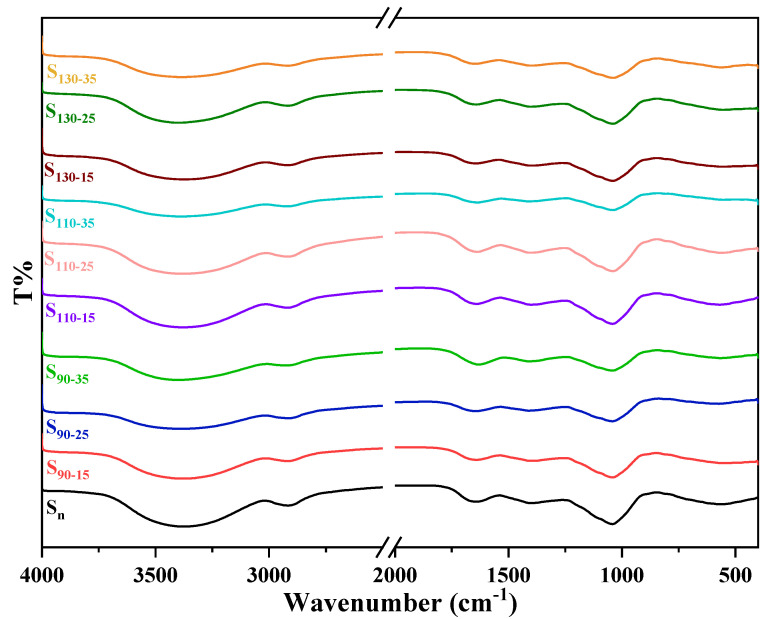
FTIR spectra of native and HMT HB flour. S_n_ indicates native HB flour; S_90−15_, S_90−25_, S_90−35_ indicate that the heating temperature of HB flour is 90 °C; S_110−15_, S_110−25_, S_110−35_ indicate that the heating temperature of HB flour is 110 °C; S_130−15_, S_130−25_, S_130−35_ indicate that the heating temperature of HB flour is 130 °C; S_90−15_, S_110−15_, S_130−15_ indicate that the moisture content of HB flour is 15%; S_90−25_, S_110−25_, S_130−25_ indicate that the moisture content of HB flour is 25%; S_90−35_, S_110−35_, S_130−35_ indicate that the moisture content of HB flour is 35%; T%, transmittance (%).

**Figure 4 foods-11-03511-f004:**
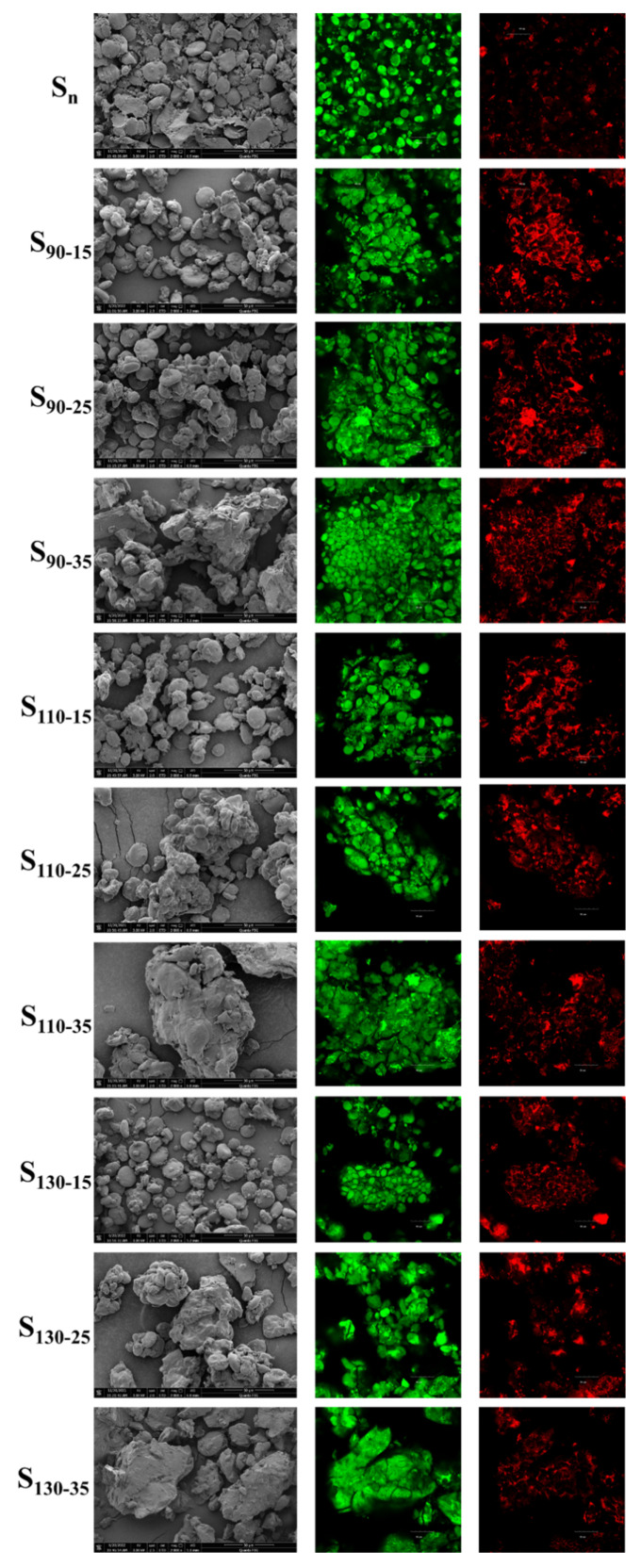
Morphological characteristics of native and HMT HB flours. S_n_ indicates native HB flour; S_90−15_, S_90−25_, S_90−35_ indicate that the heating temperature of HB flour is 90 °C; S_110−15_, S_110−25_, S_110−35_ indicate that the heating temperature of HB flour is 110 °C; S_130−15_, S_130−25_, S_130−35_ indicate that the heating temperature of HB flour is 130 °C; S_90−15_, S_110−15_, S_130−15_ indicate that the moisture content of HB flour is 15%; S_90−25_, S_110−25_, S_130−25_ indicate that the moisture content of HB flour is 25%; S_90−35_, S_110−35_, S_130−35_ indicate that the moisture content of HB flour is 35%. Starch stained by FITC emits green, protein stained by Rhodamine B emits red.

**Table 1 foods-11-03511-t001:** Pasting properties of native and HMT HB flours.

Sample	Peak Viscosity (cPa)	Trough Viscosity (cPa)	Breakdown (cPa)	Final Viscosity (cPa)	Setback (cPa)	Peak Time (s)	Pasting Temperature (°C)
S_n_	3626 ± 40 ^c^	2229 ± 7 ^c^	1397 ± 33 ^c^	3734 ± 26 ^d^	1505 ± 33 ^b^	6.32 ± 0.00 ^c^	66.93 ± 2.44 ^d^
S_90−15_	4260 ± 184 ^b^	2649 ± 116 ^b^	1611 ± 68 ^b^	4180 ± 21 ^c^	1531 ± 136 ^b^	6.25 ± 0.10 ^c^	68.15 ± 0.64 ^d^
S_90−25_	5147 ± 72 ^a^	3302 ± 1 ^a^	1845 ± 71 ^a^	5384 ± 122 ^a^	2082 ± 121 ^a^	6.35 ± 0.04 ^c^	69.68 ± 0.53 ^d^
S_90−35_	2712 ± 8 ^e^	2177± 0 ^cd^	566 ± 36 ^e^	3299 ± 55 ^e^	1153 ± 83 ^c^	6.69 ± 0.05 ^b^	88.43 ± 1.66 ^ab^
S_110−15_	4219 ± 111 ^b^	2566 ± 109 ^b^	1653 ± 2 ^b^	4515 ± 90 ^b^	1949 ± 199 ^a^	6.19 ± 0.09 ^cd^	70.13 ± 0.03 ^cd^
S_110−25_	3211 ± 3 ^d^	2228 ± 78 ^cd^	983 ± 81 ^d^	3717 ± 111 ^d^	1489 ± 189 ^b^	6.55 ± 0.14 ^b^	83.48 ± 0.53 ^ab^
S_110−35_	1408 ± 32 ^g^	1374 ± 42 ^e^	272 ± 24 ^f^	2076 ± 38 ^g^	941 ± 30 ^c^	6.98 ± 0.00 ^a^	92.00 ± 0.07 ^a^
S_130−15_	3662 ± 17 ^c^	2066 ±19 ^d^	1597 ± 2 ^b^	4062 ± 96 ^c^	1997 ± 115 ^a^	6.02 ± 0.05 ^d^	71.43 ± 0.53 ^cd^
S_130−25_	1663 ± 21 ^f^	1419 ± 42 ^e^	244 ± 21 ^f^	2313 ± 1 ^f^	895 ± 43 ^c^	6.98 ± 0.00 ^a^	80.30 ± 13.08 ^bc^
S_130−35_	471 ± 10 ^h^	-	-	912 ± 13 ^h^	-	-	93.58 ± 1.17 ^a^

Different letters in the same column indicate significant differences (*p* < 0.05). S_n_ indicates native HB flour; S_90−15_, S_90−25_, S_90−35_ indicate that the heating temperature of HB flour is 90 °C; S_110−15_, S_110−25_, S_110−35_ indicate that the heating temperature of HB flour is 110 °C; S_130−15_, S_130−25_, S_130−35_ indicate that the heating temperature of HB flour is 130 °C; S_90−15_, S_110−15_, S_130−15_ indicate that the moisture content of HB flour is 15%; S_90−25_, S_110−25_, S_130−25_ indicate that the moisture content of HB flour is 25%; S_90−35_, S_110−35_, S_130−35_ indicate that the moisture content of HB flour is 35%.

**Table 2 foods-11-03511-t002:** Thermal properties of native and HMT HB flours.

	*T_o_* (°C)	*T_p_* (°C)	*T_c_* (°C)	Δ*H* (J/g)
S_n_	55.77 ± 0.04 ^g^	60.47± 0.24 ^f^	66.16 ± 0.91 ^e^	6.41 ± 0.07 ^a^
S_90−15_	56.11 ± 0.12 ^fg^	60.62 ± 0.01 ^f^	66.98 ± 0.30 ^e^	5.24 ± 0.57 ^ab^
S_90−25_	62.54 ± 0.37 ^e^	66.66 ± 0.07 ^d^	73.47 ± 0.11 ^dc^	4.92 ± 0.05 ^bc^
S_90−35_	67.88 ± 0.39 ^c^	71.64 ± 0.13 ^c^	75.28 ± 0.28 ^d^	3.40 ± 0.63 ^d^
S_110−15_	56.80 ± 0.08 ^f^	61.98 ± 0.38 ^e^	67.42 ± 0.46 ^e^	5.58 ± 0.76 ^ab^
S_110−25_	64.09 ± 0.99 ^d^	71.46 ± 0.76 ^c^	83.07 ± 0.79 ^c^	3.91 ± 0.15 ^cd^
S_110−35_	68.02 ± 0.40 ^c^	72.67± 0.03 ^b^	85.14 ± 0.34 ^c^	1.31 ± 0.22 ^e^
S_130−15_	57.04 ± 0.17 ^f^	62.17 ± 0.46 ^e^	70.96 ± 0.64 ^c^	5.98 ± 0.38 ^ab^
S_130−25_	69.93 ± 0.11 ^b^	80.64 ± 0.80 ^a^	88.63 ± 0.12 ^b^	1.14 ± 0.40 ^e^
S_130−35_	73.72 ± 0.01 ^a^	80.69 ± 0.00 ^a^	91.71 ± 0.16 ^a^	0.43 ± 0.01 ^e^

Different letters in the same column indicate significant differences (*p* < 0.05). S_n_ indicates native HB flour; S_90−15_, S_90−25_, S_90−35_ indicate that the heating temperature of HB flour is 90 °C; S_110−15_, S_110−25_, S_110−35_ indicate that the heating temperature of HB flour is 110 °C; S_130−15_, S_130−25_, S_130−35_ indicate that the heating temperature of HB flour is 130 °C; S_90−15_, S_110−15_, S_130−15_ indicate that the moisture content of HB flour is 15%; S_90−25_, S_110−25_, S_130−25_ indicate that the moisture content of HB flour is 25%; S_90−35_, S_110−35_, S_130−35_ indicate that the moisture content of HB flour is 35%; *T_o_*, onset temperature; *T_p_*, peak temperature; *T_c_*, conclusion temperature; Δ*H*, gelatinization enthalpy.

**Table 3 foods-11-03511-t003:** RC of native and HMT HB flours.

Sample	S_n_	S_90−15_	S_90−25_	S_90−35_	S_110−15_	S_110−25_	S_110−35_	S_130−15_	S_130−25_	S_130−35_
RC (%)	28.52	30.45	41.32	32.23	36.39	36.46	31.08	36.96	29.56	31.28

S_n_ indicates native HB flour; S_90−15_, S_90−25_, S_90−35_ indicate that the heating temperature of HB flour is 90 °C; S_110−15_, S_110−25_, S_110−35_ indicate that the heating temperature of HB flour is 110 °C; S_130−15_, S_130−25_, S_130−35_ indicate that the heating temperature of HB flour is 130 °C; S_90−15_, S_110−15_, S_130−15_ indicate that the moisture content of HB flour is 15%; S_90−25_, S_110−25_, S_130−25_ indicate that the moisture content of HB flour is 25%; S_90−35_, S_110−35_, S_130−35_ indicate that the moisture content of HB flour is 35%; RC, relative crystallinity.

**Table 4 foods-11-03511-t004:** Absorbance ratio of native and HB flours.

Sample	S_n_	S_90−15_	S_90−25_	S_90−35_	S_110−15_	S_110−25_	S_110−35_	S_130−15_	S_130−25_	S_130−35_
R_1047/1022_	1.2584 ± 0.0032 ^c^	1.3245 ± 0.0062 ^abc^	1.48341 ± 0.0085 ^a^	1.5085 ± 0.0224 ^ab^	1.4599 ± 0.0159 ^abc^	1.3053 ± 0.0961 ^bc^	1.2974 ± 0.1192 ^c^	1.1997 ± 0.0929 ^c^	1.2139 ± 0.0850 ^c^	1.3579 ± 0.0289 ^abc^

Means followed by different letters in the same line are significantly different at *p* < 0.05. S_n_ indicates native HB flour; S_90−15_, S_90−25_, S_90−35_ indicate that the heating temperature of HB flour is 90 °C; S_110−15_, S_110−25_, S_110−35_ indicate that the heating temperature of HB flour is 110 °C; S_130−15_, S_130−25_, S_130−35_ indicate that the heating temperature of HB flour is 130 °C; S_90−15_, S_110−15_, S_130−15_ indicate that the moisture content of HB flour is 15%; S_90−25_, S_110−25_, S_130−25_ indicate that the moisture content of HB flour is 25%; S_90−35_, S_110−35_, S_130−35_ indicate that the moisture content of HB flour is 35%; R_1047/1022_, the bands’ absorption area ratios at 1047/1022 cm^−1^.

**Table 5 foods-11-03511-t005:** Secondary structure of protein of native and HMT HB flours.

Sample	β-Sheet Structure	Random Coil Structure	α-Helix Structure	β-Turn Structure
S_n_	0.3084 ± 0.0221 ^b^	0.1610 ± 0.0053 ^a^	0.1679 ± 0.0199 ^bc^	0.3627 ± 0.0031 ^a^
S_90−15_	0.3162 ± 0.0003 ^ab^	0.1657 ± 0.0032 ^a^	0.1929 ± 0.0034 ^ab^	0.3252 ± 0.0001 ^d^
S_90−25_	0.3216 ± 0.0017 ^ab^	0.1622 ± 0.0002 ^a^	0.1872 ± 0.0001 ^ab^	0.3291 ± 0.0020 ^cd^
S_90−35_	0.3352 ± 0.0272 ^ab^	0.1518 ± 0.0085 ^a^	0.1545 ± 0.0226 ^c^	0.3585 ± 0.0039 ^a^
S_110−15_	0.3385 ± 0.0076 ^ab^	0.1620 ± 0.0001 ^a^	0.1723 ± 0.0006 ^abc^	0.3330 ± 0.0013 ^c^
S_110−25_	0.3119 ± 0.0026 ^ab^	0.1590 ± 0.0040 ^a^	0.1863 ± 0.0009 ^ab^	0.3429 ± 0.0004 ^b^
S_110−35_	0.3196 ± 0.0055 ^ab^	0.1613 ± 0.0060 ^a^	0.1901 ± 0.0020 ^ab^	0.3290 ± 0.0015 ^cd^
S_130−15_	0.3423 ± 0.0283 ^ab^	0.1335 ± 0.0441 ^a^	0.1965 ± 0.0097 ^a^	0.3276 ± 0.0061 ^cd^
S_130−25_	0.3251 ± 0.000 ^ab^	0.1639 ± 0.0001 ^a^	0.1863 ± 0.0002 ^ab^	0.3246 ± 0.0011 ^d^
S_130−35_	0.3530 ± 0.0153 ^a^	0.1485 ± 0.0070 ^a^	0.1827 ± 0.0087 ^ab^	0.3157 ± 0.0004 ^e^

Different letters in the same column indicate significant differences (*p* < 0.05). S_n_ indicates native HB flour; S_90−15_, S_90−25_, S_90−35_ indicate that the heating temperature of HB flour is 90 °C; S_110−15_, S_110−25_, S_110−35_ indicate that the heating temperature of HB flour is 110 °C; S_130−15_, S_130−25_, S_130−35_ indicate that the heating temperature of HB flour is 130 °C; S_90−15_, S_110−15_, S_130−15_ indicate that the moisture content of HB flour is 15%; S_90−25_, S_110−25_, S_130−25_ indicate that the moisture content of HB flour is 25%; S_90−35_, S_110−35_, S_130−35_ indicate that the moisture content of HB flour is 35%.

**Table 6 foods-11-03511-t006:** RDS, SDS and RS of native and HMT HB flours.

Sample	RDS (%)	SDS (%)	RS (%)
S_n_	60.35 ± 1.28 ^e^	14.87 ± 0.22 ^d^	24.77 ± 1.50 ^bc^
S_90−15_	61.57 ± 0.25 ^e^	10.96 ± 0.00 ^e^	27.47 ± 0.25 ^b^
S_90−25_	60.33 ± 0.44 ^e^	15.89 ± 0.78 ^d^	23.78 ± 1.21 ^c^
S_90−35_	62.27 ± 0.14 ^e^	35.19 ± 0.64 ^a^	2.55 ± 0.78 ^ef^
S_110−15_	61.98 ± 0.14 ^e^	4.62 ± 0.55 ^f^	33.40 ± 0.70 ^a^
S_110−25_	65.96 ± 0.57 ^d^	15.89 ± 1.00 ^d^	18.15 ± 0.43 ^d^
S_110−35_	68.74 ± 1.09 ^c^	29.64 ± 0.52 ^b^	1.62 ± 1.61 ^ef^
S_130−15_	59.96 ± 1.52 ^e^	13.53 ± 2.13 ^de^	26.51 ± 0.60 ^bc^
S_130−25_	71.50 ± 0.70 ^b^	24.90 ± 0.66 ^c^	3.61 ± 1.36 ^e^
S_130−35_	85.30 ± 0.85 ^a^	14.12 ± 0.88 ^d^	0.58 ± 0.03 ^f^

Different letters in the same column indicate significant differences (*p* < 0.05). S_n_ indicates native HB flour; S_90−15_, S_90−25_, S_90−35_ indicate that the heating temperature of HB flour is 90 °C; S_110−15_, S_110−25_, S_110−35_ indicate that the heating temperature of HB flour is 110 °C; S_130−15_, S_130−25_, S_130−35_ indicate that the heating temperature of HB flour is 130 °C; S_90−15_, S_110−15_, S_130−15_ indicate that the moisture content of HB flour is 15%; S_90−25_, S_110−25_, S_130−25_ indicate that the moisture content of HB flour is 25%; S_90−35_, S_110−35_, S_130−35_ indicate that the moisture content of HB flour is 35%; RDS, rapidly digestible starch SDS, slowly digestible starch; RS, resistant starch.

## Data Availability

The data required to reproduce these findings cannot be shared at this time as the data also forms part of an ongoing study.

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
