# Peer review of "Effect of Heat–Moisture Treatment on the Physicochemical Properties, Structure, Morphology, and Starch Digestibility of Highland Barley (Hordeum vulgare L. var. nudum Hook. f) Flour"

_foods, 2022, doi:10.3390/foods11213511_

Round 1
Reviewer 1 Report
The manuscript deals with the synergistic effect of heat and water on the physicochemical properties, structure, morphology, and digestibility of highland barley flouron.
The manuscript is well-planned and has solid results and discussion.
However, before publication some problems have to be removed.
The abstract should present the main achievement of the paper. In that context, authors should remove text which is more appropriate to the introduction, rather than to abstract, e.g. first sentence of the abstract.
The values in tables must be presented in appropriate format and with precision related to the error bars.
Captions to Tables and Figures need modifications. Caption with Table and Figure has to be enough to understand content presented. That is not the case in a present version of the manuscript.
In general, better planning of layout of the manuscript is strongly recommended.
More work on Tables and Figures is recommended to make them more friendly to the reader.
Another problem is the English grammar and style that is not very critical, but some changes and improvements should be made. In that context, I present some suggestions to improve the conclusion part.
It is, At 15% moisture content of HB flour, no significant change was found in the morphology of the starch granules.
After changes, At 15% moisture content of HB flour, no significant changes were found in the morphology of the starch granules.
It is, HMT induced changes in starch structure such that HB starch bound easily to digestive enzymes, leading to an increase in RDS content.
After changes, HMT-induced changes in starch easily structure, such that HB starch bound to digestive enzymes, leading to an increase in RDS content.
It is, HB flour treated at low moisture content did not bind easily to digestive enzymes, leading to an increase in RS content.
After changes, HB flour treated with low moisture content could not easily bind to digestive enzymes, leading to an increase in the RS content.
Author Response
To Referee:1
Comments and Suggestions for Authors
The manuscript deals with the synergistic effect of heat and water on the physicochemical properties, structure, morphology, and digestibility of highland barley flour. The manuscript is well-planned and has solid results and discussion.
However, before publication some problems have to be removed.
Response: We feel great thanks for your professional review work on our article. We revised the problems in the paper according to your comments. We fervently hope that all revisions can be accepted.
The abstract should present the main achievement of the paper. In that context, authors should remove text which is more appropriate to the introduction, rather than to abstract, e.g. first sentence of the abstract.
Response: We feel great thanks for your friendly advice. We decided to delete the first sentence of the abstract.
The values in tables must be presented in appropriate format and with precision related to the error bars.
Response: Thank you very much for your careful review. We have corrected the incorrect formatting in Table 5 and made the values in the table consistent with the precision of the error values.
Captions to Tables and Figures need modifications. Caption with Table and Figure has to be enough to understand content presented. That is not the case in a present version of the manuscript.
Response: Thank you very much for your professional advice. We have revised the captions of the Tables and Figures and highlighted them. We hope that this will help the reader to better understand what is being presented in the Tables and Figures.
In general, better planning of layout of the manuscript is strongly recommended.
More work on Tables and Figures is recommended to make them more friendly to the reader.
Response: Thank you very much for your kind reminder and friendly suggestion. We adjusted inappropriate layout and tables of the manuscript in the article, which is of great help to our article.
Another problem is the English grammar and style that is not very critical, but some changes and improvements should be made. In that context, I present some suggestions to improve the conclusion part.
It is, at 15% moisture content of HB flour, no significant change was found in the morphology of the starch granules.
After changes, at 15% moisture content of HB flour, no significant changes were found in the morphology of the starch granules.
It is, HMT induced changes in starch structure such that HB starch bound easily to digestive enzymes, leading to an increase in RDS content.
After changes, HMT-induced changes in starch easily structure, such that HB starch bound to digestive enzymes, leading to an increase in RDS content.
It is, HB flour treated at low moisture content did not bind easily to digestive enzymes, leading to an increase in RS content.
After changes, HB flour treated with low moisture content could not easily bind to digestive enzymes, leading to an increase in the RS content.
Response: We really appreciate your suggestions for improving the language in our conclusions. Your comments on the revision of the language are very professional. We have adopted your proposed change and think it is great.
Reviewer 2 Report
Mention the scientific name of highland barley in the title, if it seems possible because there are two varieties generally named as highland barley.
Introduction is well written, but don't use the word "we" in last paragraph. Kindly change it like "in this study........"
From my point of view, 110 and 130oC temperature for HMT are too high. Results also showed that. What will be benefits of these modification for product development?
From product point of view, for which product this HMT flour will be suitable?
First line of abstract is "Highland barley (HB) has attracted much attention from scientists because of its unique nutritional composition". Nothing about nutritional importance is mentioned in introduction.
How these heat moisture treatment will affect the nutritional properties of HB?
In the conclusion, it is claimed that HMT treatment increased the RDS content. When this HMT flour will be converted into product, it will undergo heat treatment which will again change its properties.
Overall, paper is well written, well understandable provide sufficient knowledge of properties of HD flour after heat- moisture treatment but authors should also focus on product to be prepared from this flour or at least suggest that for which product, it will be suitable.
Author Response
To Referee:2
Comments and Suggestions for Authors
Mention the scientific name of highland barley in the title, if it seems possible because there are two varieties generally named as highland barley.
Response: Thank you very much for your friendly advice. We have revised the title to “Effect of heat–moisture treatment on the physicochemical properties, structure, morphology, and starch digestibility of highland barley (Hordeum vulgare L. var. nudum Hook. f) flour”.
Introduction is well written, but don't use the word "we" in last paragraph. Kindly change it like "in this study........"
Response: Thank you very much professional advice. We have deleted the word "we" from the last paragraph of the introduction. This sentence has been rewritten as:" In this study, x-ray diffraction (XRD) and Fourier-transform infrared spectroscopy (FTIR) were used to examine the structure of starch and proteins to better understand the effects of different temperatures and moisture content of HMT on HB flour".
From my point of view, 110 and 130℃ temperature for HMT are too high. Results also showed that. What will be benefits of these modifications for product development?
Response: Thank you very much this critical question. The properties of HB flour treated at 110 and 130 °C and 35% moisture content are indeed significantly different from those of HB flour treated at low moisture content or 90 °C. The rheological results show a greater tendency for S110-35 and S130-35 to be Newtonian behaviors, with increased fluidity and reduced shear thinning. Therefore, we consider that these modifications are beneficial to HB flour as an ingredient in the production of cereal beverage. During the HMT of HB flour Maillard reaction also takes place which can give the cereal beverage its unique flavour.
From product point of view, for which product this HMT flour will be suitable?
Response: Thank you very much this critical question. It has been shown that under low moisture or low temperature treatment conditions the properties of the flour change more gently, with little effect on the deterioration of the dough and the final product, and that the high content of resistant starch makes it suitable for the production of staple foods such as noodles, buns and bread. HMT has been shown to increase thermal stability and shear resistance of flour and therefore can be used as an alternative to chemically modified starches in retort foods, confections, salad dressings and batter products. This is a very critical question. We have added relevant contents in the introduction.
First line of abstract is "Highland barley (HB) has attracted much attention from scientists because of its unique nutritional composition". Nothing about nutritional importance is mentioned in introduction.
Response: Thank you very much for your kind reminder. After our consideration, we felt that it was inappropriate for this sentence to appear in the abstract and decided to remove it.
How these heat moisture treatment will affect the nutritional properties of HB?
Response: Thank you very much for raising this professional question. Heat-moisture treatment has the greatest effect on starch, as it is the most abundant component of the flour. We examined the levels of fast digesting starch, slow digesting starch and resistant starch in different HMT HB flours and found that the conditions of the HMT had different effects on the levels of the three starch types. We found that HB flour treated at 110 °C and 15% moisture had the highest resistant starch content and that consumption of this HB flour would help to slow the rate of postprandial blood glucose rise.
In the conclusion, it is claimed that HMT treatment increased the RDS content. When this HMT flour will be converted into product, it will undergo heat treatment which will again change its properties.
Response: Thank you very much for raising this professional and interesting question. We very much agree with this view. Reheating HMT flour after it is made into products will really change their properties. However, due to different kinds of foods, their heating methods are also different. Common food heating methods include steam, boiling, baking, frying and microwave. Different heating methods have different effects. Steam heating causes degradation of starch macromolecules and breakdown of starch granules during pasting, resulting in elevated levels of rapidly digestible starch. On the contrary, the baking not only makes the starch less gelatinization, but also increases the content of resistant starch.
Overall, paper is well written, well understandable provide sufficient knowledge of properties of HD flour after heat- moisture treatment but authors should also focus on product to be prepared from this flour or at least suggest that for which product, it will be suitable.
Response: Thank you very much for your kind reminder and friendly suggestion. We have added relevant contents in the introduction. Your comments can help us to improve the quality of our paper.
Reviewer 3 Report
The topic is interesting and manuscript well organized.
Comments for improvement has been indicated within the manuscript.
Additionally, the pictures quality need to be enhanced.

Author Response
To Referee:3
Comments and Suggestions for Authors
The topic is interesting and manuscript well organized.
Comments for improvement has been indicated within the manuscript.
Additionally, the pictures quality need to be enhanced.
Response: Thank you for your careful review and friendly advice. We have revised the manuscript in accordance with the attached annotations and responded one by one.
Include type of digestibility.
Response: Thank you for your professional advice. We have revised the title to “Effect of heat–moisture treatment on the physicochemical properties, structure, morphology, and starch digestibility of highland barley (Hordeum vulgare L. var. nudum Hook. f) flour”.
Please Define
Response: Thank you for your careful review. ΔH represents the gelatinization enthalpy. We have used gelatinization enthalpy instead of ΔH in the abstract.
Check font
Response: Thank you for your careful review. We have revised "Fourier-transform infrared spectroscopy" to the correct font.
Define on first use
Response: Thank you for your kind reminder. We have supplemented the definition of “KBr” where it was first used
Kindly describe how the conditioning was done
Response: We feel great thanks for your professional review work on our article. We have rewritten this sentence as “An appropriate amount of distilled water was sprayed evenly and slowly on the surface of the HB flour, and constant stirring was carried out to avoid uneven mixing. The moisture content of the HB flour was adjusted to 15%, 25%, and 35%, respectively, and then equilibrated at a temperature of 4°C for 24 h in sealed polyethylene bags”.
Add space between value and ℃ here and every other place used
Response: Thank you for your kind reminder. We have added a space between the value and °C. Your comments were very helpful for our paper.
What type of bag?
Response: Thank you for your careful review. The material of the sealed bags we use is polyethylene and we have added a note on the material of the bags in the manuscript.
Define the codes
Response: Thank you very much for your kind reminder. Sn represents native HB flour, A and B in SA-B represent the temperature and moisture content of the HMT conditions respectively.
Add place of manufacture
Response: Thank you for your careful review and friendly advice. The HAAKE RS6000 rheometer was produced in Germany, as we have added in the manuscript.
You may also add discussion of result based on statistical differences (significance level)
Response: Thank you very much for your professional suggestion. We have added discussion of result based on statistical.
Why this citation. What did this author do similar to yours.
Response: Thank you for your careful review. An oversight in our work has led to irregularities in citation. We have removed this reference. Thank you again for your careful review.
Please remove full stop in the titles of tables an figures
Response: Thank you very much for your kind reminder and friendly suggestion. We have removed the full stop from the titles of tables and figures in the manuscript.
Look into this. There was an exception of decrease compared to untreated sample. What could be responsible.
Response: Thank you for your careful review. We checked our experimental data again and found that we had entered the wrong value, the correct value should be 66.98 ± 0.30.
Sounds too affirmative. You may adopt the word “could”
Response: Thank you very much for your kind reminder and friendly suggestion. We have rewritten the sentence according to your suggestion as “The increase in To, Tp, and Tc of HB flour after HMT could be due to the interactions between amylose and amylose, amylose and amylopectin, and amylose and fat molecules, which limited the flexibility of the internal amorphous structure of starch and hindered its swelling”.
Please define
Response: Thank you very much for your professional suggestion. We have added the definition of tan δ in the manuscript.
The figure is not clear as the colours were not well contrasted. Use better picture.
Title of figure should be under.
Response: Thank you very much for your professional suggestion. We have revised the figure in the manuscript and improved its clarity.
This table is missing
Response: Thank you for your careful review. We are very sorry for we ignore the Table 6. We have added Table 6 to the manuscript.
Which of the treatment in each analysis seen to produce better/best result compared with untreated sample or other samples. Please amplify.
Response: Thank you very much for your professional and friendly advice, which helped us a lot to improve our paper. Our experimental results found that the effect of different treatment conditions on HB flour was different. All conditions of HMT improved the thermal stability of HB flour. At low moisture content or 90 °C, HMT significantly increases the viscosity of HB flour. HMT significantly increases the viscosity and relative crystallinity of HB flour at low moisture content or at 90 °C. Furthermore, the HB flour treated at 15% moisture content also had an increased content of resistant starch, which was a noteworthy result. We have added the relevant content and highlighted it in the conclusion.
Add an statement on general contribution to knowledge.
Response: We really appreciate your comments on the article. Your comments have been very helpful in improving the quality of our paper. We have added statement to the conclusion.
Round 2
Reviewer 1 Report
The manuscript has been improved; however, still it is a problem with numbers in Table 1, for example instead 3626 ± 39.60c it should be 3626 ± 40.
in all tables where symbols are used, clear explanation of the symbol should be presented, bellow the table.
In Figures still caption improvement is needed. Authors should clearly explain what relation is presented in Figure, referring to the symbols used.
Author Response
To Referee:1
Comments and Suggestions for Authors
The manuscript has been improved; however, still it is a problem with numbers in Table 1, for example instead 3626 ± 39.60c it should be 3626 ± 40.
Response: Thank you very much for your affirmation of the revision work of the paper. Your comments are very important for the improvement of the quality of our paper. We have revised the numbers in Table 1 in response to your comments.
in all tables where symbols are used, clear explanation of the symbol should be presented, bellow the table.
Response: Thanks again for your careful review. We have added explanations of what the symbols stand for below all the tables. We hope this revision will be accepted.
In Figures still caption improvement is needed. Authors should clearly explain what relation is presented in Figure, referring to the symbols used.
Response: We feel great thanks for your professional review work on our article. Because of our mistake, we didn't revise our manuscript correctly last time. We have revised the problems in Figure 2 and Figure 3. And explanations of the symbols have been added below all the figures.
Thank you very much again for your very patient and careful review of our manuscript.